# Efficient Information-Theoretic Large-Scale Semi-Supervised Metric Learning via Proxies

Peng Chen [1] and Huibing Wang [2,*]

1   Drilling and Production Technology Research Institute, Liaohe Oilfield Company, PetroChina, Panjin 124010, China; peng_chen@petrochina.com.cn
2   College of Information and Science Technology , Dalian Maritime University, Dalian 116021, China
*   Correspondence: huibing.wang@dlmu.edu.cn

**Abstract:** Semi-supervised metric learning intends to learn a distance function from the limited labeled data as well as a large amount of unlabeled data to better gauge the similarities of any two instances than using a general distance function. However, most existing semi-supervised metric learning methods rely on the manifold assumptions to mine the rich discriminant information of the unlabeled data, which breaks the intrinsic connection between the manifold regularizer-building process and the subsequent metric learning. Moreover, these methods usually encounter high computational or memory overhead. To solve these issues, we develop a novel method entitled Information-Theoretic Large-Scale Semi-Supervised Metric Learning via Proxies (ISMLP). ISMLP aims to simultaneously learn multiple proxy vectors as well as a Mahalanobis matrix and forms the semi-supervised metric learning as the probability distribution optimization parameterized by the Mahalanobis distance between the instance and each proxy vector. ISMLP maximizes the entropy of the labeled data and minimizes that of the unlabeled data to follow the entropy regularization, in this way, the labeled part and unlabeled part can be integrated in a meaningful way. Furthermore, the time complexity of the proposed method has a linear dependency concerning the number of instances, thereby, can be extended to the large-scale dataset without incurring too much time. Experiments on multiple datasets demonstrate the superiority of the proposed method over the compared methods used in the experiments.

**Keywords:** semi-supervised metric learning; entropy regularization; image retrieval; Riemannian optimization

## 1. Introduction

Distance Metric Learning (DML), usually referring to learning a Mahalanobis matrix from the given side information, has been an active studying field for the last two decades [1–4]. Compared to those off-the-shelf distance functions, e.g., Euclidean distance, DML takes the correlations and weights of the features into distance consideration, thus being more appropriate for various downstream tasks. Its efficiency has been validated by a large spectrum of applications [5–7], for example, few-shot learning [8,9], face recognition [9,10], and fault detection [11,12]. Despite the success of existing DML methods, they rely on massive side information constructed by labeled information [13]. However, manually labeling the data is a labor-consuming task [14], and sometimes it needs domain knowledge [15,16] to provide meaningful labeling information, e.g., labeling the checkup samples of patients.

To solve this issue, researchers have devoted themselves to Semi-Supervised Distance Metric Learning (SSDML). SSDML intends to learn a Mahalanobis matrix from limited labeled data as well as a large amount of unlabeled data, such that under this metric, similar instances are brought closer together whereas dissimilar ones are pushed farther away. Inspired by the unsupervised dimensionality reduction methods, which aim to

preserve some properties in the original data space, a lot of SSDML approaches based on manifold-based regularization terms have been proposed in the last decades [1,17–22]. For example, Wang [1] proposed to project the data into a new space, where labeled data has the maximum margin constraint, and the unlabeled data has the maximum variance. Similarly, Baghshah and Shouraki [18] constructed a novel SDML method by retaining locally linear relationships between close data points in the transformed space and proposed a regularization term based on Locally Linear Embedding (LLE) [23]. The above regularization terms cannot boost the discriminative ability of the model. There are also some SSDML methods based on the Laplacian graph [19,21,22,24–27], i.e., Laplacian Regularized Metric Learning (LRML) [17] utilized the graph Laplacian to preserve the neighbor relationship of the original space. However, the above graph Laplacian construction process does not take the labeled information into consideration. To mitigate this issue, Dutta and Sekhar [21] proposed to utilize Markov random walk technology to transform the strong limited labeled information into a Laplacian matrix. Ying et al. [20] also took the density information of each instance into the Laplacian graph construction process. However, these methods rely on a default metric to determine the affinities among the samples, which contradicts the goal of metric learning. If the default metric is an appropriate metric, why should we still strive to search for another metric? There are also a few works that do not depend on manifold-based regularization. For example, Semi-Supervised Metric Learning Paradigm with Hyper Sparsity (SERAPH) [28] and Semi-Supervised Regularized Large Margin Distance Metric Learning (S-RLMM) [29]. SERAPH is an information-theoretic metric learning method, and it maximizes the entropy on the labeled data while minimizing the entropy on the unlabeled data. However, the time complexity of these methods is at least quadratically dependent on the number of training instances (Table 4 provides a brief time complexity analysis of some representative methods), which means these methods can hardly scale to large-scale datasets. Moreover, these methods rely on a fixed metric to mine the information similarities between samples, which contradicts the goal of learning a metric from data.

To solve this issue, in this paper, we propose an efficient SSDML method called ISMLP; rather than building the probability model via the instance–instance distance parameterized by the learned Mahalanobis matrix, we propose to learn a set of proxy vectors and transform the instance–instance relationship as the instance–proxy relationship. We minimize the labeled instances and their corresponding proxy vectors to efficiently mine the information of the unlabeled data; inspired by the SERAPH, we incorporate entropy regularization. Importantly, the Mahalanobis matrix is constricted as a hierarchical form to further boost training efficiency. An Alternating Direction Method (ADM) technology is adopted to seek a feasible solution for ISMLP, and the sub-problem concerning the Mahalanobis matrix can be efficiently solved by an iterative method on the product space of two Riemannian manifold. The merits of using proxy vectors lie in two folds: on the one hand, the time complexity of ISMLP is linearly dependent on the number of instances, thus can be easily extended to large-scale datasets; on the other hand, the instance–instance distances may be corrupted because of the noise instances in the dataset. The proxy vectors can be considered as aggregating the class/local information of the dataset, therefore, it is more stable than SERAPH.

- We propose a novel information-theoretic-based SSDML method called ISMLP, which simultaneously learns multiple proxy vectors as well as the Mahalanobis matrix. Specifically, we adopt the entropy regularization to mine the discriminant information of the unlabeled data.
- The merits of the proposed ISMLP lie in two folds: on the one hand, compared to those manifold-based SSDML methods, ISMLP does not rely on manifold assumptions. Thus, it can be applied to border scenes; the time complexity of ISMLP is linear with respect to the number of training instances, and thus can be easily extended to large-scale datasets.

- Extensive experimental results on classification and retrieval experiments can validate the superiority performance and in the meantime can be trained more efficiently than those compared methods.

The rest of this paper is organized as follows: In Section 2, we briefly introduce the SERAPH framework. Then, we introduce the construction process of the proposed method in Section 3, followed by the extensive numerical experiment in Section 5. Finally, in Section 6, we make a conclusion and provide a possible future direction of the proposed ISMLP.

## 2. Related Work

*SERAPH Framework*

Recently, Niu et al. proposed a semi-supervised metric learning framework called SERAPH, based on entropy regularization [28]. Given the probability distribution parameterized by the Mahalanobis distance between two instances, SERAPH maximizes the entropy of the sample pairs from the similar set and minimizes the entropy of those from the dissimilar set. The objective function of SERAPH can be constructed as follows:

$$
\begin{aligned}
\max_{\mathbf{A}\in\mathcal{S}_+^d} \sum_{(x_i,x_i)\in\mathcal{P}} &\log \hat{p}_{ij}^{\mathbf{A}}(y_{ij}) - \\
\mu \sum_{(x_i,x_i)\in\mathcal{U}} \sum_{y\in\{-1,1\}} &\hat{p}_{ij}^{\mathbf{A}}(y)\log \hat{p}_{ij}^{\mathbf{A}}(y) - \lambda \mathrm{Tr}(\mathbf{A}),
\end{aligned}
\tag{1}
$$

where $\lambda > 0$ and $\mu > 0$ are two hyperparameters. $\mathcal{P} = \mathcal{S} \cup \mathcal{D}$ with $\mathcal{S}(\mathcal{D})$ denoting the similar (dissimilar) set, which is defined in Section 3.1. $\mathcal{U} = \{(x_i,x_j) \mid (x_i,x_j) \notin \mathcal{P}\}$. The trace regularization ensures $\mathbf{A}$ to be low-rank. $y_{ij}$ ($y$) denotes the ground truth (predicted) label of $(x_i,x_j)$, more specifically when $(x_i,x_j) \in \mathcal{S}$, $y_{ij} = 1$ when $(x_i,x_j) \in \mathcal{D}$, $y_{ij} = -1$. $p(\hat{y}_{ij})$ represents the predicted probability of a pair of examples $(x_i,x_j)$ given the Mahalanobis matrix $\mathbf{A}$, which is defined as:

$$
p(\hat{y}_{ij}) = \frac{1}{1 + \exp\left(y_{ij}\left(d_{\mathbf{A}}^2(x_i,x_j) - \eta\right)\right)},
\tag{2}
$$

where $\eta > 0$ denotes the margin hyper-parameter.

## 3. Information-Theoretic Large-Scale Semi-Supervised Metric Learning via Proxies

In this section, we first provide the detailed construction procedure of the proposed ISMLP method. Then, we derive the optimization strategy of ISMLP.

### 3.1. Notations and Problem Definition

Given a dataset, $\mathbf{X} = [\mathbf{X}_l, \mathbf{X}_u] \in \mathbb{R}^{d\times n}$ composed of the labeled data $\mathbf{X}_l \in \mathbb{R}^{d\times n_l}$ and the unlabeled dataset $\mathbf{X}_u \in \mathbb{R}^{d\times n_u}$, where $n_l$ and $n_u$ denote the number of labeled instances and unlabeled instances, respectively, in the semi-supervised setting, $n_l$ is usually far smaller than $n_u$. $d$ is the dimensionality of the feature. For the labeled dataset $\mathbf{X}_l$, suppose that each instance is associated with the class label, i.e., $\mathbf{X}_l = \{(x_1,y_1),(x_2,y_2),\cdots,(x_{n_l},y_{n_l})\}$, with $y \in \{0,1\}^C$, $C$ is the number of the classes. Pairwise constraint sets $\mathcal{S}$ and $\mathcal{D}$ can be extracted from $\mathbf{X}_l$:

$$
\mathcal{S} = \{(x_i,x_j) \mid x_i \text{ and } x_j \text{ are semantic similar}\}
$$
$$
\mathcal{D} = \{(x_i,x_j) \mid x_i \text{ and } x_j \text{ are semantic dissimilar}\}.
$$

Semi-supervised metric learning aims to learn a Mahalanobis matrix $\mathbf{M}$ from $\mathbf{X}$, such that data can be transformed into a new space, where the semantic similarity between data points can be diametrically estimated from their distances in the transformed space.

A typical formulation of metric learning is to require the pairwise distance from $\mathcal{S}$ to be smaller than $l$ and that of the dissimilar set to be larger than $u$.

$$
\begin{aligned}
d_{\mathbf{M}}^2(x_i, x_j) &< l, \ \forall (x_i, x_j) \in \mathcal{S} \\
d_{\mathbf{M}}^2(x_i, x_k) &> u, \ \forall (x_i, x_k) \in \mathcal{D}
\end{aligned}
\tag{3}
$$

The general form of a semi-supervised metric learning framework can be constructed as:

$$
\min_{\mathbf{M}} \ell^l(\mathbf{M}, \mathcal{S}, \mathcal{D}) + \lambda \ell^u(\mathbf{M}, \mathbf{X}_u) + \mu \mathcal{R}(\mathbf{M}),
\tag{4}
$$

where $\ell^l$ and $\ell^u$ denote the pairwise loss and the unlabeled loss, respectively. $\mathcal{R}$ is the regularization term concerning the structural information of the metric. $\lambda$ and $\mu$ are two hyper-parameters that control the weight of the $\ell^u$ and $\mathcal{R}$.

*3.2. Learning From Proxy Vectors*

SEPAPH approach relies on the pairwise distance to construct the probability model, which has the following two drawbacks: (1) the time complexity of SERAPH has a quadratic dependence on the number of unlabeled data, which means it cannot scale to large-scale datasets; (2) SERAPH is a global metric learning method and SERAPH is sensitivity to the outlier in the dataset. To solve these problems, suppose that there are multiple proxy vectors anchoring around the whole instance space, and these vectors can be expressed as a set: $\mathcal{Z} = \{z_1, z_2, \cdots, z_m\}$, where $m$ is the number of vectors. The function of the proxy vector can be considered as the mean center of each class, or anchor that aggregates the local similarity information of some local instances [30]. By aligning the relevant instances to their corresponding proxy vectors, the problem of the outlier instances inside the dataset as well as the high time complexity can be efficiently settled. Similar to proxy-NCA [31], for any instance $x$, its corresponding proxy vector can be estimated by:

$$
p(x) = \arg\min_{p \in \mathcal{Z}} d_{\mathbf{M}}^2(x, z),
\tag{5}
$$

where $d_{\mathbf{M}}^2(x, z)$ is the Mahalanobis distance between $x$ and $z$. Equation (5) means that the proxy vector of an instance is the proxy vector that is closest to the instance under the metric $\mathbf{M}$.

Instead of directly constructing the probabilistic model from the pairwise Mahalanobis distance, we utilize the pairwise distances among the instance and all the proxy vectors to form the NCA-style [32] probabilistic model. More specifically, for any instance, $x_i$, its probability of randomly choosing a proxy vector can be estimated via:

$$
q_{ij} = \frac{\exp\left(-d_{\mathbf{M}}^2(x_i, z_j)\right)}{\sum_{j=1}^m \exp\left(-d_{\mathbf{M}}^2(x_i, z_j)\right)}.
\tag{6}
$$

$q_{ij}$ can reflect the closeness between $x_i$ and $z_j$; the closer $x_i$ to $z_j$, the bigger the value. All the $q_{ij}$ $(j = 1, 2, \cdots, m)$ form a valid discrete probability distribution:

$$
q_i = (q_{i1}, q_{i2}, \cdots, q_{im})^T.
\tag{7}
$$

Suppose that the weakly-supervised information is provided in the form of pairwise constraint forms. For any two instances $(x_i, x_j)$ from $\mathcal{S}$, we aim to minimize the distance between $x_i$ and the proxy vector of $x_j$, and, in the meanwhile, keep a large distance from the other proxy vectors. Such an idea can be expressed as:

$$
\max_{\mathbf{M} \in \mathcal{S}_+^d} \frac{1}{|\mathcal{S}|} \sum_{(x_i, x_j) \in \mathcal{S}} \log\left(\frac{\exp\left(-d_{\mathbf{M}}^2(x_i, p(x_j))\right)}{\sum_{j=1}^m \exp\left(-d_{\mathbf{M}}^2(x_i, z_j)\right)}\right).
\tag{8}
$$

where $|\mathcal{S}|$ denotes the cardinality of $\mathcal{S}$. $\mathcal{S}_+^d$ is the set of all the $d \times d$ PSD (Positive Semi-Definite) matrices. Maximizing Equation (8) will push $x_i$ toward $p(x_j)$. In contrast to the SERAPH algorithm, ISMLP converts the pairwise distance to that of the instance and the corresponding proxy vector. The proxy vector can be viewed as aggregating the local similarity information of local instances. Therefore, the proposed method can exhibit more robust behavior than SERAPH.

### 3.3. Entropy Regularization

The above Equation (8) only considers the labeled data information. In semi-supervised settings, the amount of labeled data is usually limited, applying Equation (8) to these conditions may easily come across the over-fitting problem. To solve this issue, researchers propose to simultaneously mine the discriminant information of unlabeled data. Multiple tricks have been taken in the literature, such as manifold-based regularization [20,21,33] and entropy regularization [28]. However, these regularizations are usually of high time or space complexity. By using the proxy vector, we can show that the time complexity of the entropy regularization can be significantly reduced.

In the field of information theory, the information entropy measures the degree of "uncertainty" of the given random variable. For a given random variable $\bar{p} = (p_1, \cdots, p_m)$, its definition of the information entropy can be expressed as $H_m(\bar{p}) = -\sum_{i=1}^m p_m \log p_m$. When $p_1 = \cdots = p_{i-1} = p_{i+1} = \cdots = p_m = 0, p_i = 1, \forall i = 1, \cdots, m$, $H_m(\bar{p})$ reaches its minimal value. In this case, the system has minimal "uncertainty" [34].

The intuition of entropy regularization used in semi-supervised learning follows the low-density separation assumption [35,36], which encourages the unlabeled data to be predicted with high probability. In ISMLP, it is desirable for unlabeled data to locate around a certain proxy vector. In other words, the distribution in Equation (7) should be a perky one. To achieve this goal, according to the above analysis of the minimization of information entropy, the following entropy regularization can be constructed:

$$\min_{\mathbf{M} \in \mathcal{S}_+^d} -\frac{1}{n_u} \sum_{i=n_l+1}^n \sum_{k=1}^m q_{ik} \log q_{ik}, \tag{9}$$

where $q_{ij}$ is the probability of $i$-th instance choosing the $j$-th proxy-vector as a neighborhood, and it is defined by Equation (6). Minimizing Equation (9) will shrink the distances between the unlabeled instances and their corresponding proxy vectors while keeping a large distance to the other proxy vectors.

### 3.4. Joint Dimensional Reduction and Metric Learning

One can combine the labeled part and the entropy regularization to finish the objective function of ISMLP. However, when the dimensionality of the feature is large, adopting PGD technology to solve ISMLP will encounter high time complexity. To solve this issue, inspired by the hierarchical way to build the Mahalanobis matrix [37], we propose to decompose $\mathbf{M}$ as the following form:

$$\mathbf{M} = \mathbf{P}\mathbf{R}\mathbf{P}^T, \tag{10}$$

where $\mathbf{P} \in \text{St}(p, d)$, and $\mathbf{R} \in \mathcal{S}_{++}^p$. In the experiment, $p$ is usually much smaller than $d$; therefore, the running time can be significantly reduced. The decomposition form can be understood as first projecting the original feature into a lower embedding space, then using $\mathbf{R}$ to learn the weight and correlations in the embedding space. Following the common practice in dimensionality reduction, we require $\mathbf{P}$ to be a column-orthogonal matrix, namely $\mathbf{P}^T\mathbf{P} = \mathbf{I}_p$.

Integrating the Equations (8)–(10) to finish the objective function of ISMLP, we obtain:

$$\min_{\mathbf{P},\mathbf{R},\mathcal{Z}} -\frac{1}{|\mathcal{S}|} \sum_{(x_i,x_j)\in\mathcal{S}} \log\left(\frac{\exp\left(-d_{\mathbf{M}}^2(x_i,p(x_j))\right)}{\sum_{j=1}^m \exp\left(-d_{\mathbf{M}}^2(x_i,z_j)\right)}\right)$$
$$-\frac{\lambda}{n_u}\sum_{i=n_l+1}^n\sum_{k=1}^m q_{ik}\log q_{ik} + \mu r(\mathbf{R},\mathbf{R}_0) \tag{11}$$
$$\text{s.t. } \mathbf{M} = \mathbf{PRP}^T, \mathbf{P} \in \mathrm{St}(p,d), \ \mathbf{R} \in \mathcal{S}_{++}^p$$

where $r : \mathcal{S}_{++}^p \times \mathcal{S}_{++}^p \to \mathbb{R}_+$ denotes the regularization term concerning $\mathbf{R}$. Here, we aim to keep $\mathbf{R}$ to be close to a prior matrix $\mathbf{R}_0$ and propose to utilize the Burg divergences [38]:

$$r(\mathbf{R},\mathbf{R}_0) = \mathrm{Tr}\left(\mathbf{RR}_0^{-1}\right) - \mathrm{logdet}\left(\mathbf{RR}_0^{-1}\right) - p. \tag{12}$$

In our experiments, we set $\mathbf{R}_0$ as $\mathbf{I}_p$ for simplicity.

$\lambda > 0$ and $\mu > 0$ are two hyper-parameters that control the importance of the entropy regularization term and the structural prior of $\mathbf{M}$, respectively. $p$ is the dimensionality of the latent space. The proposed ISMLP jointly learns the decomposition forms of the Mahalanobis matrix $(\mathbf{P},\mathbf{R})$ and multiple proxy vectors $\mathcal{Z}$. Maximizing Equation (11) will push the labeled data moving toward their corresponding proxy vectors, which will assign the unlabeled data a high probability.

Since the first part can be considered as minimizing the entropy of the labeled data and maximizing that of the unlabeled data. These two parts can be naturally combined together in a meaningful way. Instead of directly building the probabilistic model via the pairwise Mahalanobis distance as SERAPH did, ISMLP takes advantage of the proxy vectors to convert the semi-supervised metric learning into $m$ class distribution optimization. The merits of the ISMLP lie in two folds; on the one hand, the proposed IMSLP is more robust than SERAPH, and can cope well with the outlier in the dataset; on the other hand, the time complexity of the algorithm can be significantly reduced.

## 4. Optimization for ISMLP

There are three types of parameters to be estimated in the objective function of ISMLP. In this section, an alternating-direction technology is proposed to seek a feasible solution. More specifically, we keep the other variables fixed to update the current variable until the stop condition meet.

**Fix $\mathcal{Z}$, to solve P and R**: the sub-problem concerning $\mathbf{P}$ and $\mathbf{R}$ can be expressed as the following Riemannian manifold-based optimization problem.

$$\min_{\mathbf{P},\mathbf{R}} \mathcal{F}(\mathbf{P},\mathbf{R}|\mathbf{X},\mathbf{M}_0)$$
$$\text{s.t. } \mathbf{P} \in \mathrm{St}(p,d), \ \mathbf{R} \in \mathcal{S}_{++}^p. \tag{13}$$

The above minimization problem can be solved via the product space of the Stiefel and SPD manifold. According to [39], the Stiefel and SPD manifolds are locally homogeneous spaces, and their product space should also follow the smoothness and differentiability. Therefore, $\mathcal{M}_p = \mathrm{St}(p,d) \times \mathcal{S}_{++}^p$ can contain a Riemannian structural.

**Theorem 1.** *The set $\left(\mathrm{St}(p,d) \times \mathcal{S}_{++}^p\right)\backslash\mathcal{O}(p)$ with the following equivalence relation*

$$[(\mathbf{P},\mathbf{R})] \sim \left\{\left(\mathbf{PQ},\mathbf{Q}^T\mathbf{RQ}\right), \forall\mathbf{Q} \in \mathcal{O}(p)\right\} \tag{14}$$

*and Riemannian metric*

$$g_{(\mathbf{P},\mathbf{R})}((\xi_{\mathbf{P}},\xi_{\mathbf{R}}),(\zeta_{\mathbf{P}},\zeta_{\mathbf{R}})) = 2\mathrm{Tr}\left(\xi_{\mathbf{P}}^T\zeta_{\mathbf{P}}\right) +$$
$$\mathrm{Tr}\left(\mathbf{R}^{-1}\xi_{\mathbf{R}}\mathbf{R}^{-1}\zeta_{\mathbf{R}}\right) \tag{15}$$

*forms a Riemannian quotient manifold.*

**Proof.** We first prove that the equivalence relation hold: $\mathcal{F}(\mathbf{P}, \mathbf{R}|\mathbf{X}, \mathbf{M}_0) = \mathcal{F}(\mathbf{PQ}, \mathbf{Q}^T\mathbf{RQ}|\mathbf{X}, \mathbf{M}_0)$, since the following equation holds:

$$\mathbf{PQQ}^T\mathbf{RQ}(\mathbf{PQ})^T = \mathbf{PRP}^T, \forall \mathbf{Q} \in \mathcal{O}(p). \tag{16}$$

Therefore, the equivalence relation holds. To prove $\mathcal{M}_p\backslash\mathcal{O}(p)$ is a valid quotient manifold, one can follow the proof in [40]. Lastly, as for the Riemannian metric, interested readers can refer to [37].

To perform the Riemannian gradient descent on $\mathcal{M}_p$, we usually follow the "projection and retraction" procedures. More specifically, firstly, one can transform the Euclidean gradient as the Riemannian gradient, and then perform the gradient descent step; then, map the intermediate solution back to the manifold [39]. For the Stiefel manifold, the Riemannian gradient can be computed as:

$$\xi_{\mathbf{P}} = \frac{\partial \mathcal{F}}{\partial \mathbf{P}} - \frac{1}{2}\mathbf{P}\left(\mathbf{P}^T\frac{\partial \mathcal{F}}{\partial \mathbf{P}} + \left(\frac{\partial \mathcal{F}}{\partial \mathbf{P}}\right)^T\mathbf{P}\right). \tag{17}$$

As for the PSD manifold, its Riemannian gradient has the following form:

$$\xi_{\mathbf{R}} = \frac{1}{2}\mathbf{R}\left(\frac{\partial \mathcal{F}}{\partial \mathbf{R}} + \left(\frac{\partial \mathcal{F}}{\partial \mathbf{M}}\right)^T\right)\mathbf{R}, \tag{18}$$

where $\frac{\partial \mathcal{F}}{\partial \mathbf{P}}$ and $\frac{\partial \mathcal{F}}{\partial \mathbf{R}}$ denote the Euclidean partial gradient of $\mathcal{F}$ w.r.t $\mathbf{P}$ and $\mathbf{R}$, respectively.

As for the quotient manifold $\mathcal{M}_p$, the tangent space at $\Gamma = (\mathbf{P}, \mathbf{R})$ is divided into two complementary parts, namely a horizontal part $\mathcal{H}_\Gamma\mathcal{M}_p$ and a vertical one $\mathcal{V}_\Gamma\mathcal{M}_p$. Importantly, the tangent space of $\mathcal{M}_p$ (denoted as $\mathcal{T}_\Gamma\mathcal{M}_p$) can be uniquely identified as it horizontal part.

The horizontal vector in the horizontal tangent space of the proposed quotient manifold can be identified as:

$$(\xi_{\mathbf{P}} - \mathbf{P}\psi, \xi_{\mathbf{R}} - \mathbf{R}\psi + \psi\mathbf{R}), \tag{19}$$

where $\psi$ is the solution of the following equation [37]:

$$\psi\mathbf{R}^2 + \mathbf{R}^2\psi = \mathbf{R}\left(\xi_{\mathbf{P}}^T\mathbf{P} - \mathbf{P}^T\xi_{\mathbf{P}} + \mathbf{R}^{-1}\xi_{\mathbf{R}} - \xi_{\mathbf{R}}\mathbf{R}^{-1}\right)\mathbf{R}. \tag{20}$$

For the retraction operation, it can be organized in the following form:

$$\mathcal{R}_{(\mathbf{P},\mathbf{R})}(\xi_{\mathbf{P}}, \xi_{\mathbf{R}}) = \left(\mathrm{uf}(\mathbf{P} + \xi_{\mathbf{P}}),\right.$$
$$\left.\mathbf{R}^{\frac{1}{2}}\left(\exp\left(\mathbf{R}^{-\frac{1}{2}}\xi_{\mathbf{R}}\mathbf{R}^{-\frac{1}{2}}\right)\right)\right)\mathbf{R}^{\frac{1}{2}}, \tag{21}$$

where $\mathrm{uf}(\mathbf{B}) = \mathbf{B}(\mathbf{B}^T\mathbf{B})^{-\frac{1}{2}}$, and $\exp(\cdot)$ denotes the matrix exponential operation.

Lastly, the only missing component is $\frac{\partial \mathcal{F}}{\partial \mathbf{P}}$ and $\frac{\partial \mathcal{F}}{\partial \mathbf{R}}$, and they can be calculated by:

$$\frac{\partial \mathcal{F}}{\partial \mathbf{P}} = \frac{1}{|\mathcal{S}|}\sum_{(x_i, x_j)\in\mathcal{S}}\left(\sum_{k=1}^m q_{ik}\frac{\partial d_{\mathbf{M}}^2(x_i, z_j)}{\partial \mathbf{P}} - \frac{\partial d_{\mathbf{M}}^2(x_i, p(x_j))}{\partial \mathbf{P}}\right)$$
$$- \frac{\lambda}{n_u}\sum_{i=n_l+1}^n\sum_{k=1}^m(1 + \log q_{ik})q_{ik}\left(-\frac{\partial d_{\mathbf{M}}^2(x_i, z_k)}{\partial \mathbf{P}}\right.$$
$$\left.+ \sum_{j=1}^m \exp\left(-d_{\mathbf{M}}^2(x_i, z_j)\right)\frac{\partial d_{\mathbf{M}}^2(x_i, z_k)}{\partial \mathbf{P}}\right), \tag{22}$$

with $\dfrac{\partial d_{\mathbf{M}}^{2}(\boldsymbol{x}_i, \boldsymbol{z}_k)}{\partial \mathbf{P}} = 2(\boldsymbol{x}_i - \boldsymbol{z}_k)(\boldsymbol{x}_i - \boldsymbol{z}_k)^{T}\mathbf{P}\mathbf{R}.$

For $\dfrac{\partial \mathcal{F}}{\partial \mathbf{R}}$, it can be expressed as:

$$
\begin{aligned}
\frac{\partial \mathcal{F}}{\partial \mathbf{R}} = {} & \frac{1}{|\mathcal{S}|} \sum_{(\boldsymbol{x}_i, \boldsymbol{x}_j) \in \mathcal{S}} \left( \sum_{k=1}^{m} q_{ik} \frac{\partial d_{\mathbf{M}}^{2}(\boldsymbol{x}_i, \boldsymbol{z}_j)}{\partial \mathbf{R}} - \frac{\partial d_{\mathbf{M}}^{2}(\boldsymbol{x}_i, p(\boldsymbol{x}_j))}{\partial \mathbf{R}} \right) \\
& - \frac{\lambda}{n_u} \sum_{i=n_l+1}^{n} \sum_{k=1}^{m} (1 + \log q_{ik}) q_{ik} \left( - \frac{\partial d_{\mathbf{M}}^{2}(\boldsymbol{x}_i, \boldsymbol{z}_k)}{\partial \mathbf{R}} \right. \\
& + \left. \sum_{j=1}^{m} \exp\left( -d_{\mathbf{M}}^{2}(\boldsymbol{x}_i, \boldsymbol{z}_j) \right) \frac{\partial d_{\mathbf{M}}^{2}(\boldsymbol{x}_i, \boldsymbol{z}_k)}{\partial \mathbf{R}} \right) + \mu \left( \mathbf{R}_0^{-1} - \mathbf{R}^{-1} \right),
\end{aligned}
\tag{23}
$$

with $\dfrac{\partial d_{\mathbf{M}}^{2}(\boldsymbol{x}_i, \boldsymbol{z}_k)}{\partial \mathbf{R}} = \mathbf{P}^{T}(\boldsymbol{x}_i - \boldsymbol{z}_k)(\boldsymbol{x}_i - \boldsymbol{z}_k)^{T}\mathbf{P}.$

**Fix P and R to solve $\mathcal{Z}$:** The sub-problem with respect to $\mathcal{Z}$ can be stated as:

$$
\begin{aligned}
\min \mathcal{G}(\mathcal{Z}) = {} & -\frac{1}{|\mathcal{S}|} \sum_{(\boldsymbol{x}_i, \boldsymbol{x}_j) \in \mathcal{S}} \log \left( \frac{\exp\left( -d_{\mathbf{M}}^{2}(\boldsymbol{x}_i, p(\boldsymbol{x}_j)) \right)}{\sum_{j=1}^{m} \exp\left( -d_{\mathbf{M}}^{2}(\boldsymbol{x}_i, \boldsymbol{z}_j) \right)} \right) \\
& + \frac{\lambda}{n_u} \sum_{i=n_l+1}^{n} \sum_{k=1}^{m} q_{ik} \log q_{ik}.
\end{aligned}
\tag{24}
$$

Firstly, we can update the proxy assignment of each instance by recalculating Equation (5). Then, we can solve the proxy vector one by one. More specifically, for the $k$-th proxy vector $\boldsymbol{z}_k$, by taking the derivative of $\mathcal{G}$ with respect to $\boldsymbol{z}_k$, we can get:

$$
\begin{aligned}
\frac{\partial \mathcal{G}}{\partial \boldsymbol{z}_k} = {} & \frac{1}{|\mathcal{S}|} \left( \sum_{\substack{(\boldsymbol{x}_i, \boldsymbol{x}_j) \in \mathcal{S} \\ p(\boldsymbol{x}_j) \neq \boldsymbol{z}_k}} \frac{1}{q_{ip(\boldsymbol{x}_j)}} \frac{\partial q_{ip(\boldsymbol{x}_j)}}{\partial \boldsymbol{z}_k} + \sum_{\substack{(\boldsymbol{x}_i, \boldsymbol{x}_j) \in \mathcal{S} \\ p(\boldsymbol{x}_j) = \boldsymbol{z}_k}} \frac{1}{q_{ik}} \frac{\partial q_{ik}}{\partial \boldsymbol{z}_k} \right) \\
& - \frac{\lambda}{n_u} \sum_{i=n_l+1}^{n} \left( \sum_{l \neq k} \left( (1 + \log q_{il}) \frac{\partial q_{il}}{\partial \boldsymbol{z}_k} \right) + (1 + \log q_{ik}) \frac{\partial q_{ik}}{\partial \boldsymbol{z}_k} \right)
\end{aligned}
\tag{25}
$$

where for the given similar pair $(\boldsymbol{x}_i, \boldsymbol{x}_j)$, $q_{ip(\boldsymbol{x}_i)}$ denotes the probability of $\boldsymbol{x}_i$ choosing $p(\boldsymbol{x}_j)$ as the proxy vector computed by Equation (6). $\frac{\partial q_{ik}}{\partial \boldsymbol{z}_k} = 2(1 - q_{ik})\mathbf{M}(\boldsymbol{x}_i - \boldsymbol{z}_k)$, and $\frac{\partial q_{il}}{\partial \boldsymbol{z}_k} = \frac{-2q_{il}}{\sum_{j=1}^{m} \exp\left( -d_{\mathbf{M}}^{2}(\boldsymbol{x}_i, \boldsymbol{z}_j) \right)}\mathbf{M}(\boldsymbol{x}_i - \boldsymbol{z}_k).$

By setting $\frac{\partial \mathcal{G}}{\partial \boldsymbol{z}_k}$ to zero, Equation (25) is simply a linear equation:

$$
w\mathbf{M}\boldsymbol{z}_i = \boldsymbol{\psi}
\tag{26}
$$

where

$$
\begin{aligned}
w = {} & \frac{1}{|\mathcal{S}|} \left( \sum_{\substack{(\boldsymbol{x}_i, \boldsymbol{x}_j) \in \mathcal{S} \\ p(\boldsymbol{x}_j) \neq \boldsymbol{z}_k}} \frac{1}{\sum_{j=1}^{m} \exp\left( -d_{\mathbf{M}}^{2}(\boldsymbol{x}_i, \boldsymbol{z}_j) \right)} - \right. \\
& \sum_{\substack{(\boldsymbol{x}_i, \boldsymbol{x}_j) \in \mathcal{S} \\ p(\boldsymbol{x}_j) = \boldsymbol{z}_k}} \frac{1 - q_{ik}}{q_{ik}} \right) - \frac{\lambda}{n_u} \left( \sum_{i=n_l+1}^{n} \left( 1 + \log q_{ik} \right) \frac{1 - q_{ik}}{q_{ik}} \right. \\
& + \left. \sum_{l \neq k} (1 + \log q_{il}) \frac{q_{il}}{\sum_{j=1}^{m} \exp\left( -d_{\mathbf{M}}^{2}(\boldsymbol{x}_i, \boldsymbol{z}_j) \right)} \right),
\end{aligned}
\tag{27}
$$

and $\psi$:

$$
\begin{aligned}
\psi = \frac{\mathbf{M}}{|\mathcal{S}|} \Bigg( & \sum_{\substack{(\boldsymbol{x}_i, \boldsymbol{x}_j) \in \mathcal{S} \\ p(\boldsymbol{x}_j) \neq \boldsymbol{z}_k}} \frac{-\boldsymbol{x}_i}{\sum_{j=1}^{m} \exp\left(-d_{\mathbf{M}}^2\left(\boldsymbol{x}_i, \boldsymbol{z}_j\right)\right)} + \\
& \sum_{\substack{(\boldsymbol{x}_i, \boldsymbol{x}_j) \in \mathcal{S} \\ p(\boldsymbol{x}_j) = \boldsymbol{z}_k}} \frac{1 - q_{ik}}{q_{ik}} \boldsymbol{x}_i \Bigg) + \frac{\lambda}{n_u} \left( \sum_{i=n_l+1}^{n} \left(1 + \log q_{ik}\right) \frac{1 - q_{ik}}{q_{ik}} \right. \\
& \left. - \sum_{l \neq k} (1 + \log q_{il}) \frac{q_{il}}{\sum_{j=1}^{m} \exp\left(-d_{\mathbf{M}}^2\left(\boldsymbol{x}_i, \boldsymbol{z}_j\right)\right)} \right) \mathbf{M}\boldsymbol{x}_i,
\end{aligned}
\tag{28}
$$

we can get the closed-form solution of $\boldsymbol{z}_k$:

$$
\boldsymbol{z}_k = (\mathbf{M} + \eta \mathbf{I}_d)^{-1} \frac{\psi}{w}
\tag{29}
$$

where $\eta > 0$ is a small positive number to make $\mathbf{A}$ a positively defined matrix. In the experiment, we empirically set it as $1e^{-6}$, which works fine . □

To sum up, we propose an alternating direction strategy to solve the minimization of Equation (11). The sub-problems concerning $\mathbf{P}$ and $\mathbf{R}$ are updated on the product manifold of the Stiefel and SPD manifold via the Riemannian gradient descent algorithm [41]. The sub-problem concerning $\mathcal{Z}$ has a closed-form solution. The main procedure of ISMLP is documented in Algorithm 1. It should be noted that we utilize a Gaussian Mixture Model (GMM) to initialize the set of proxy vectors which are the means of the corresponding components. We also document the variations of objective function values with respect to iterations on three datasets in Figure 1. Clearly, the loss decreases as the iterations and inclines become stable after several iterations, which proves that the proposed algorithm can converge within limited iterations.

---

**Algorithm 1:** The optimization strategy of ISMLP.

---

**Input:** The labeled and unlabeled datasets $\mathbf{X}_l \in \mathbb{R}^{d \times n_l}$ and $\mathbf{X}_u \in \mathbb{R}^{d \times n_u}$, the similar constraint set $\mathcal{S}$, $\lambda$, the number of proxy vectors $m$, $\mu$, and the dimensionality of the embedding space p;

1　Initialize $\mathbf{P} \in \mathbb{R}^{d \times p}$ as a column orthonormal matrix, and setting $\mathbf{R}$ as the identity matrix $\mathbf{I}_p$;

2　Initialize the set of proxy vectors via the Gaussian mixture model by setting the number of components as $m$;

3　**while** *not converged* **do**

4　　Fix $\mathcal{Z}$ to solve $\mathbf{P}$ and $\mathbf{R}$ via the Riemannian gradient descent algorithm by using the Equations (19) and (21);

5　　**for** *k = 1, ..., m* **do**

6　　　Fix $\mathbf{P}$ and $\mathbf{R}$ to solve the *k*-th proxy vector $\boldsymbol{z}_k$ by Equation (29);

7　　**end**

8　**end**

**Output:** The projection matrix $\mathbf{P}$ and low-dimensionality Mahalanobis matrix $\mathbf{R}$;

---

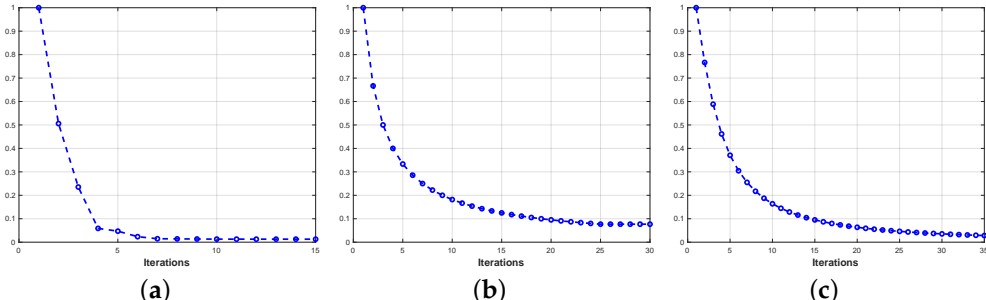

**Figure 1.** The variations of the normalized objective function values of the proposed method with the number of iterations on MNIST, Corel 5K, and Cars-196. Obviously, the loss value decreases with the number of iterations and inclines to become stable after several iterations. (**a**) MNIST; (**b**) Corel 5K; (**c**) Cars-196.

*Time Complexity of ISMLP*

In this section, we provide a brief analysis of the time complexity of the proposed ISMLP. Recall that the number of labeled and unlabeled data is $n_l$ and $n_u$, respectively. The dimensionality of the original and the reduced data is denoted as $d$ and $p$. The number of proxy vectors is denoted as $m$. The main procedures of ISMLP consist of the following main steps: (1) Evaluating the loss function; (2) Computing the Euclidean gradient of the loss function with respect to $\mathbf{P}$ and $\mathbf{R}$; (3) Projecting the Euclidean gradient of $\mathbf{P}$ and $\mathbf{R}$ to the tangent space and then retracing them back to the manifold; (4) Updating the set of the proxy vectors.

- Since the time complexity of solving the inverse of a $p \times p$ matrix costs $\mathcal{O}(p^3)$, evaluating the objective function in Equation (11) takes $\mathcal{O}(npd + mnp^2 + p^3)$.
- Computing the Euclidean gradient of the loss function with respect to $\mathbf{P}$ by using Equation (22) takes $\mathcal{O}(mndp^2)$, and computing $\frac{\partial \mathcal{F}}{\partial \mathbf{R}}$ via Equation (23) costs $\mathcal{O}(mnp^2 + p^3)$.
- According to [37], projecting the Euclidean gradient of the loss function with respect to $\mathbf{P}$ and $\mathbf{R}$ by using Equations (17) and (18) costs $\mathcal{O}(4dp^2 + 3p^3)$. Retracting the Riemannian gradient back to the manifold via Equation (21) costs $\mathcal{O}(4dp^2 + 14p^3)$.
- Solving all the proxy vectors by using Equation (29) cost $\mathcal{O}(d^3 + m^2 n_l p^2)$, where $n_l$ is the number of labeled data.

Considering the usual case of large-scale semi-supervised metric learning is $d \ll n$, the time complexity of the proposed ISMLP is about $\mathcal{O}(mndp^2)$, and the main cost lies in evaluating the Euclidean gradient of $\mathbf{P}$. To sum up, the time complexity of the proposed ISMLP has a linear dependence on the number of $n$; therefore, it can be effectively and efficiently extended to large-scale datasets.

## 5. Experiment

In this section, extensive visual classification and retrieval experiments are conducted to verify the efficacy and efficiency of the proposed ISMLP. Firstly, we provide a detailed description of the datasets and evaluation index, and compared methods used in the experiments. Then, the experimental results are provided.

### 5.1. Datasets, Evaluation Protocol and Compared Methods

**Datasets:** a total of five datasets are utilized, including MNIST [42], Fashion-MNIST [43], Corel 5K [44], CUB-200 [45], and Cars-196 [46]. The MNIST contains 70,000 grayscale handwritten digital images from ten classes, whereas Fashion-MNIST consists of 70,000 images from ten fashion objects. MNIST and Fashion-MNIST are wildly used in the field of semi-supervised learning. The latter two datasets CUB-200 and Cars-196 are two fine-grained visual recognition datasets. The detailed information on the datasets is listed in Table 1.

**Table 1.** The detailed information of datasets used in the experiment.

| Dataset | Type | Class | Instance | Feature | Train, Validation, Test |
|---|---|---|---|---|---|
| MNIST | Image | 10 | 70,000 | 784 | 50,000, 10,000, 10,000 |
| Fashion-MNIST | Image | 10 | 70,000 | 784 | 50,000, 10,000, 10,000 |
| Corel 5K | Image | 50 | 5000 | 2048 | 4000, 500, 500 |
| CUB-200 | Image | 200 | 11,788 | 2048 | 4994, 1000, 5794 |
| Cars-196 | Image | 196 | 16,185 | 2048 | 7144, 1000, 8041 |

The pixel value of MNIST and Fashion-MNIST datasets are served as the image feature, which provides us with a 784-dimensional feature. As for the other datasets, the VGG-19 network [47] pre-trained on ImageNet is utilized to extract the features. Since the dimensionality of the features is extremely high, PCA is adopted to reduce the dimensionality of each feature to a 150-dimensional subspace.

**Evaluation protocol:** Given that each dataset comes with a default partition of training/testing set, we adopt the same strategy for consistency. Additionally, for each dataset, we set aside 1000 instances from the training data to form the validation set (Since Corel 5K has a default partition of the validation set, we exclude it). The specific number of instances used for validation can be found in Table 1. For the MNIST and Fashion-MNIST datasets, we adopt 3-nearest neighbors to quantify the performance of each compared method, whereas, for the CUB-200 and Cars-196 datasets, we report the Recall@K (abbreviation as R@K) performance, where R@K reflects the proportion of the correct samples in the return $K$ samples. More specifically, R@1, R@2, R@4, and R@8 index is utilized to measure the performance of each method.

We report the R@K of each method under different labeling rates, namely 5%, 10%, and 30%, the rest samples in the training set serve as the unlabeled data.

**Compared methods:** We compare the proposed ISMLP with several state-of-the-art semi-supervised metric learning methods including, LSeMML [22], SERAPH [28], S-RLMM [29], LRML [33], SLRML [19], APIT [21], CMM [1], and APLLR [21]. One supervised metric learning method LMNN [48], one deep semi-supervised metric learning entitled as SSML-DR [49], the Euclidean distance denoted as EUCLID is also adopted for a baseline method. The hyper-parameters of all the methods are tuned on the validation set and we choose those parameters that achieve the best results on the validation set. For example, for LMNN, we tune $\lambda$ from the set $\{0.1, 0.2, \cdots, 0.9\}$. As for the proposed method, we empirically set $p$ as 50, and choose $\mu$ from the range $\{0.00001, 0.0001, \cdots, 1000\}$, and tune the $\lambda$ from the range $\{0.0001, 0.001, \cdots, 100, 1000\}$, the number for the proxy vector is chosen from $\{\#Class, 2\#Class, 3\#Class\}$. For SSML-DR, to make a fair comparison, a three-layer full-connected neural network whose nodes are $\{128, 256, 128\}$ is incorporated as the backbone network by SSML-DR. The batch size is set as 100, and the number of epochs is set as 50.

### 5.2. Classification Experimental Result on MNIST and Fashion-MNIST Datasets

In this section, we test the classification performance of the compared methods based on 3-nearest neighbors on MNIST and Fashion-MNIST datasets. To mitigate the influence of the random partition of the dataset, we repeat each task 30 times, and the mean accuracy and standard deviation are recorded to quantify the performance of each method. Tables 2 and 3 record the mean error rate and standard deviation of all the methods on MNIST and Fashion-MNIST datasets, respectively.

It is readily seen that the metric learning methods can boost the performance of $k$-nearest neighbors classification, and all the methods can benefit from the amount of labeled data. The performance of the supervised metric learning method LMNN shows inferior performance compared to those semi-supervised methods, which can prove the necessity of utilizing the information of unlabeled data during the metric learning process. Both SERAPH and ISMLP utilize entropy regularization to preserve the discriminative information of the unlabeled data. Unlike SERAPH, ISMLP adopts a set of proxy vectors to

substitute the sample–sample probability assigning procedure, which is more robust than SERAPH. As a result, its performance surpasses SERAPH on all the tasks. Compared to those Laplacian-graph-based methods (i.e., LSeMML, SLRML, and APIT), ISMLP is free from the untrustable Laplacian-graph construction process; thus, it usually can achieve better performance. ISMLP obtains the best performance on 5/6 tasks. It is curious to see that CMM achieves the worst performance among all the semi-supervised methods; we surmise its manifold-based regularization term may cause this. CMM intends to find a projection direction where the unlabeled data has the maximum variance, which may not increase the discriminative ability of the model. Owing to the powerful nonlinear feature extraction ability of the deep neural network, the performance of SSML-DR consistently surpasses those shallow Laplacian graph-based methods; however, it still falls behind the proposed ISMLP. We believe that this can be attributed to the two-stage construction processes of the Laplacian graph.

To systematically provide a comprehensive analysis of the time complexity of each method, we provide the time complexity of some representative methods in Table 4. Clearly, the time complexity of the proposed ISMLP is linear with respect to the number of instances, whereas the other compared methods exhibit at least quadratic dependence on $n$. Considering the usual case in the large-scale semi-supervised setting is $d \ll n$, the proposed ISMLP can be efficiently trained in a reasonable time. To verify this hypothesis, we also conduct experiments on MNIST and Fashion-MNIST to compare the training time of each method. Figure 2 displays the results, and clearly, the training time of ISMLP is significantly less than the compared methods. More specifically, on the MNIST dataset, it takes about 400 s for SLP to train the model, a 6.5× improvement over the second-fastest approach CMM, which can prove the efficiency of the proposed ISMLP.

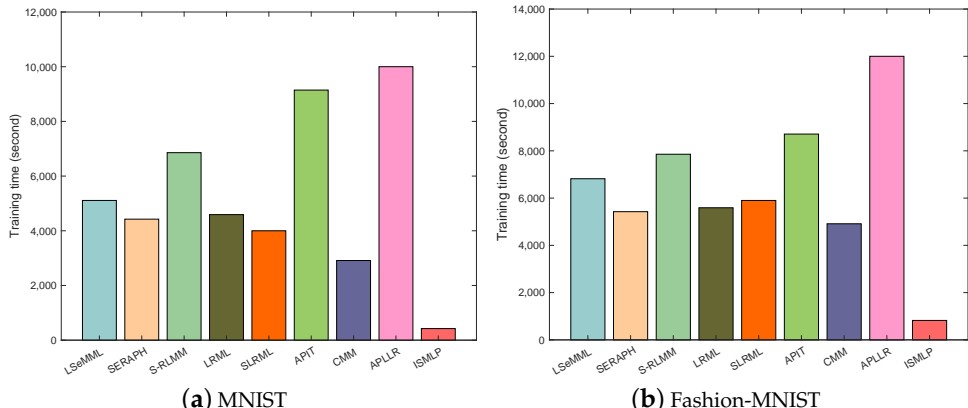

(**a**) MNIST          (**b**) Fashion-MNIST

**Figure 2.** The training time of different methods on two image datasets: (**a**) displays the training time of different methods on MNIST dataset; (**b**) shows the result on Fashion-MNIST dataset.

**Table 2.** Classification performance (mean error rate, and standard deviation) of all the compared methods based on 3-NN with varying labeling rates on MNIST dataset; the best performance of each task is marked with boldface.

| | EUCLID | LSeMML | SERAPH | S-RLMM | LRML | SLRML | APIT | CMM | APLLR | LMNN | SSML-DR | ISMLP |
|---|---|---|---|---|---|---|---|---|---|---|---|---|
| 5% labeled data | $0.296 \pm 0.005$ | $0.228 \pm 0.010$ | $0.231 \pm 0.014$ | $0.241 \pm 0.027$ | $0.221 \pm 0.011$ | $0.208 \pm 0.008$ | $0.228 \pm 0.010$ | $0.245 \pm 0.021$ | $0.209 \pm 0.022$ | $0.261 \pm 0.022$ | $0.200 \pm 0.013$ | $0.211 \pm 0.018$ |
| 10% labeled data | $0.240 \pm 0.007$ | $0.191 \pm 0.013$ | $0.184 \pm 0.011$ | $0.211 \pm 0.016$ | $0.201 \pm 0.010$ | $0.221 \pm 0.015$ | $0.198 \pm 0.011$ | $0.227 \pm 0.017$ | $0.189 \pm 0.009$ | $0.231 \pm 0.021$ | $0.175 \pm 0.012$ | $\mathbf{0.170 \pm 0.015}$ |
| 30% labeled data | $0.186 \pm 0.006$ | $0.131 \pm 0.012$ | $0.140 \pm 0.007$ | $0.127 \pm 0.014$ | $0.153 \pm 0.012$ | $0.130 \pm 0.009$ | $0.146 \pm 0.008$ | $0.150 \pm 0.016$ | $0.128 \pm 0.007$ | $0.143 \pm 0.010$ | $0.120 \pm 0.016$ | $\mathbf{0.116 \pm 0.011}$ |

**Table 3.** Classification performance (mean error rate, and standard deviation) of all the compared methods based on 3-NN with varying labeling rate on Fashion-MNIST dataset; the best performance of each task is marked with boldface.

| | EUCLID | LSeMML | SERAPH | S-RLMM | LRML | SLRML | APIT | CMM | APLLR | LMNN | SSML-DR | ISMLP |
|---|---|---|---|---|---|---|---|---|---|---|---|---|
| 5% labeled data | $0.355 \pm 0.008$ | $0.281 \pm 0.017$ | $0.289 \pm 0.014$ | $0.291 \pm 0.017$ | $0.295 \pm 0.012$ | $0.278 \pm 0.008$ | $0.298 \pm 0.012$ | $0.302 \pm 0.012$ | $0.285 \pm 0.013$ | $0.292 \pm 0.019$ | $0.248 \pm 0.010$ | $\mathbf{0.252 \pm 0.014}$ |
| 10% labeled data | $0.281 \pm 0.009$ | $0.242 \pm 0.012$ | $0.248 \pm 0.011$ | $0.237 \pm 0.012$ | $0.241 \pm 0.017$ | $0.239 \pm 0.007$ | $0.247 \pm 0.013$ | $0.258 \pm 0.016$ | $0.227 \pm 0.012$ | $0.261 \pm 0.010$ | $0.218 \pm 0.009$ | $\mathbf{0.210 \pm 0.012}$ |
| 30% labeled data | $0.235 \pm 0.006$ | $0.180 \pm 0.014$ | $0.172 \pm 0.009$ | $0.178 \pm 0.012$ | $0.172 \pm 0.011$ | $0.181 \pm 0.008$ | $0.187 \pm 0.011$ | $0.191 \pm 0.013$ | $0.188 \pm 0.012$ | $0.192 \pm 0.010$ | $0.168 \pm 0.011$ | $\mathbf{0.162 \pm 0.012}$ |

**Table 4.** The time complexity of several typical semi-supervised learning methods, where $|\mathcal{P}| = |\mathcal{S}| + |\mathcal{D}|$, $r$ is the number of iterations in each method, and $c$ (in the APID) denotes the number of the inner iterations. Clearly, the time complexity of the proposed ISMLP has a linear dependence on the $n$.

| | LSeMML | SERAPH | S-RLMM | LRML | SLRML | APIT | CMM | ISMLP |
|---|---|---|---|---|---|---|---|---|
| Time complexity | $\mathcal{O}\left(n^2 \log n + |\mathcal{P}|d^2\right)$ | $\mathcal{O}\left((n^2 d + d^3)r\right)$ | $\mathcal{O}\left((n^2 d^2 + d^3)r\right)$ | $\mathcal{O}\left(n^2 \log n + d^3\right)$ | $\mathcal{O}\left(n^2 d + d^3\right)$ | $\mathcal{O}\left(n^3 + cd^2\right)$ | $\mathcal{O}\left((n^2 + |\mathcal{P}|d^2)\right)$ | $\mathcal{O}\left((mndp^2 + d^3)\right)$ |

*5.3. Retrieval Performance on Corel 5K Dataset*

We also run a retrieval experiment on Corel 5K dataset with a labeling rate of 30%. Figure 3 documents the performance of the proposed ISMLP and LSeMML. In the first sub-figure, it is evident that ISMLP can effectively capture the semantic meaning of the query image, leading to accurate retrieval of five relevant images. In contrast, LSeMML mistakes the "red" element as the key property of the query image and unsurprisingly gives the irrelevant images in the 4-th and 5-th nearest neighbors. The major difference between ISMLP and LSeMML lies in the utilization of the unlabeled regularization term, i.e., LSeMML utilizes the EUCLID metric to mine the manifold information of the unlabeled data. When the EUCLID metric is not appropriate to measure the correlations and weights of the features, the resulting graph will be an inferior one. Therefore, the sub-optimal results can be observed in the retrieval list. In contrast, the proposed ISMLP utilizes entropy regularization to mine the information of the unlabeled data, it makes no data distribution assumption; therefore, can be applied to broader scenes.

Similar results can also be found in the other sub-figures. Therefore, the retrieval experiment on Corel 5K dataset can verify the superiority performance over the compared LSeMML approach.

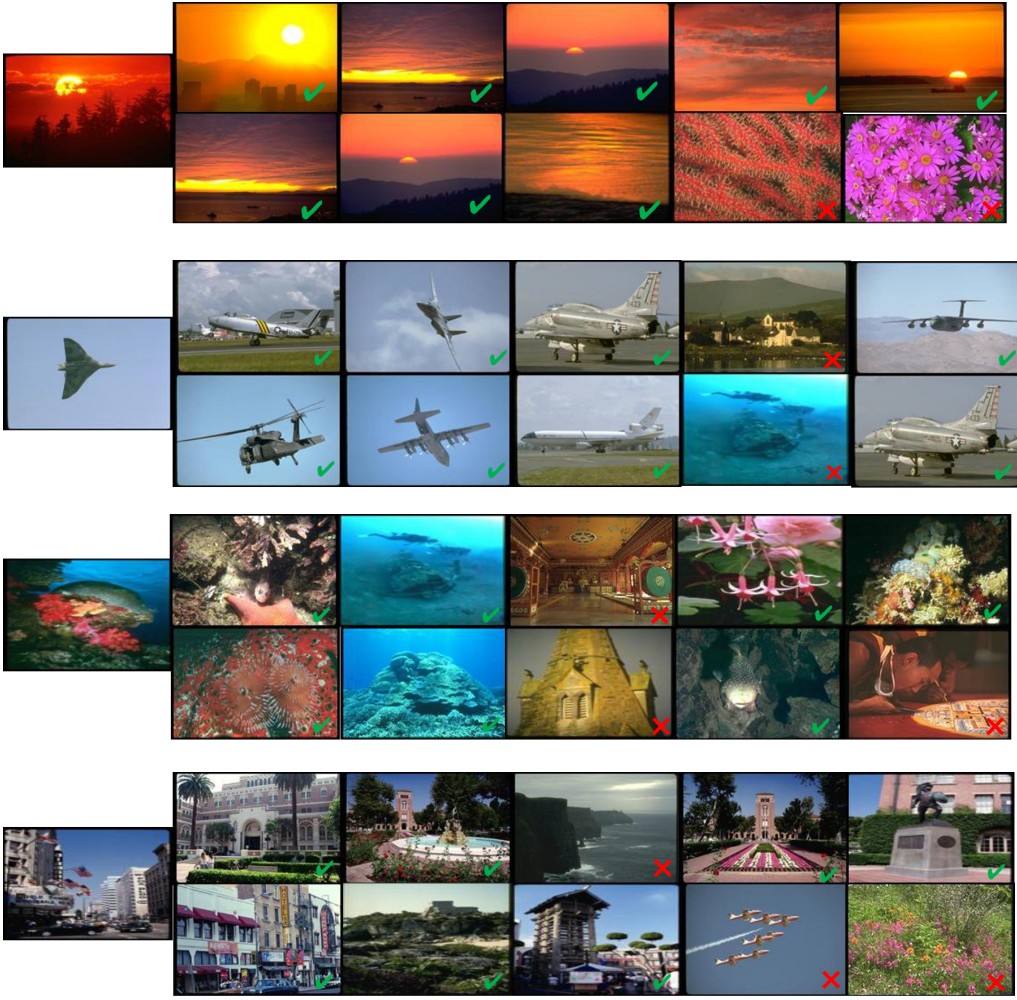

**Figure 3.** The typical retrieval result of the proposed ISMLP and LSeMML approach on the Corel 5K dataset with a training labeling rate of 30%. The leftmost figure denotes the query image and the first row of each sub-figure displays the result of the proposed method, whereas the second row shows the result of LSeMML. The green checkmark means the right retrieval result while the red cross means the wrong results.

## 5.4. Classification Performance on CUB-200 and Cars-196 Datasets

We further conduct image recognition experiments on two fine-grained datasets to test the classification ability of the proposed ISMLP and its compared methods. The R@K index is utilized to quantify the performance of each method. Tables 5 and 6 show the classification results on the CUB-200 and Cars-196 datasets, respectively.

We can draw the following conclusions from the figure: (1) All metric learning methods can benefit from the amount of labeled data; the more labeled data, the higher recognition performance. Since LMNN can only utilize the labeled data, when the labeling rate is low, LMNN can easily fall into the trap of over-fitting. Its performance is inferior to those semi-supervised metric learning methods under all tasks. This can prove the superiority of utilizing the unlabeled in metric learning. (2) Compared to those manifold-based semi-supervised approaches, the proposed ISMLP makes no assumptions about the smoothness or density of the data. Thus, ISMLP can be applied to broader scenes, and achieve better performance. (3) Both SERAPH and ISMLP utilize entropy regularization to mine the discrimination information of the unlabeled data; ISMLP adopts the proxy vectors to construct the probability model, which is more robust than SERAPH, and it obtains better performance across all the tasks on CUB-200 and Cars-196 datasets. (4) SSML-DR can obtain competing results due to its strong hierarchical feature extraction ability. (5) the proposed ISMLP can better mine the rich structural information of the unlabeled data; it achieves the best performance on 20 of all 24 tasks, which proves the efficiency of adopting the proxy vectors as surrogate points.

**Table 5.** The performance of the proposed ISMLP and compared methods on the CUB-200 dataset with varying labeling rates. The best performance under each index is marked in bold.

| | 5%Labeled Data | | | | 10%Labeled Data | | | | 30%Labeled Data | | | |
|---|---|---|---|---|---|---|---|---|---|---|---|---|
| | **R@1** | **R@2** | **R@4** | **R@8** | **R@1** | **R@2** | **R@4** | **R@8** | **R@1** | **R@2** | **R@4** | **R@8** |
| EUCLID | 25.75 | 29.82 | 32.73 | 34.82 | 26.85 | 32.57 | 34.12 | 36.45 | 29.68 | 32.22 | 36.90 | 39.29 |
| LSeMML | 32.84 | 34.54 | 36.90 | 38.73 | 34.53 | 36.72 | 38.13 | 40.65 | 35.90 | 37.81 | 39.72 | 42.20 |
| SERAPH | 33.11 | 35.61 | 37.81 | 39.98 | 35.02 | 36.87 | 38.42 | 41.97 | 37.83 | 39.24 | 42.69 | 44.73 |
| S-RLMM | 32.83 | 34.81 | 36.31 | 38.71 | 34.63 | 36.59 | 38.20 | 40.21 | 38.80 | 40.68 | 42.98 | 43.73 |
| LRML | 31.10 | 33.68 | 36.83 | 38.50 | 33.76 | 35.80 | 37.81 | 39.69 | 37.33 | 39.84 | 42.38 | 44.16 |
| SLRML | 33.63 | 34.19 | 37.24 | 39.42 | 35.82 | 37.67 | 39.29 | 42.19 | 37.90 | 39.19 | 40.57 | 42.78 |
| APIT | 32.52 | 34.57 | 37.99 | 38.68 | 34.68 | 36.85 | 39.09 | 36.98 | 37.81 | 40.83 | 42.68 | 44.73 |
| CMM | 32.13 | 34.11 | 36.73 | 38.10 | 34.82 | 37.40 | 37.49 | 39.80 | 37.86 | 40.13 | 41.68 | 42.68 |
| APLLR | 31.16 | 33.29 | 35.73 | 37.96 | 33.24 | 36.76 | 39.85 | 42.48 | 36.66 | 39.83 | 42.08 | 45.71 |
| LMNN | 29.71 | 32.90 | 35.84 | 37.80 | 31.84 | 33.59 | 37.85 | 39.84 | 33.83 | 35.49 | 39.90 | 41.83 |
| SSML-DR | 34.00 | 35.95 | 38.34 | 40.50 | 36.84 | 39.00 | 43.19 | 45.54 | 38.26 | 42.18 | 45.40 | 48.21 |
| ISMLP | **34.42** | **36.82** | **38.90** | **41.70** | **36.99** | **39.24** | **44.45** | **46.02** | **39.84** | **42.80** | **45.61** | **48.90** |

**Table 6.** The performance of the proposed ISMLP and compared methods on the Cars-196 dataset with varying labeling rates. The best performance under each index is marked in bold.

| | 5%Labeled Data | | | | 10%Labeled Data | | | | 30%Labeled Data | | | |
|---|---|---|---|---|---|---|---|---|---|---|---|---|
| | **R@1** | **R@2** | **R@4** | **R@8** | **R@1** | **R@2** | **R@4** | **R@8** | **R@1** | **R@2** | **R@4** | **R@8** |
| EUCLID | 24.63 | 28.16 | 31.34 | 34.73 | 25.68 | 30.65 | 33.87 | 35.71 | 28.56 | 31.89 | 34.17 | 38.81 |
| LSeMML | 33.26 | 34.54 | 37.15 | 39.27 | 34.58 | 37.18 | 39.46 | 42.17 | 36.19 | 39.43 | 41.87 | 44.98 |
| SERAPH | 32.19 | 34.61 | 36.25 | 39.89 | 35.18 | 37.26 | 39.48 | 43.16 | 37.30 | 39.56 | 42.43 | 45.25 |
| S-RLMM | 34.37 | 36.81 | 37.18 | 39.96 | 36.58 | 38.78 | 40.41 | 44.34 | 38.48 | 41.68 | 44.15 | 47.28 |
| LRML | 32.82 | 35.33 | 36.83 | 37.41 | 33.72 | 35.41 | 37.19 | 39.71 | 38.18 | 41.29 | 43.21 | 45.87 |
| SLRML | 34.37 | 36.19 | 38.98 | 40.57 | 35.42 | 38.42 | 40.76 | 43.87 | 37.57 | 39.58 | 42.81 | 46.10 |
| APIT | 33.46 | 35.72 | 37.99 | 39.51 | 34.71 | 36.78 | 39.18 | 41.57 | 38.24 | 41.58 | 43.79 | 46.28 |
| CMM | 34.19 | 35.45 | 36.73 | 39.10 | 35.88 | 38.04 | 40.00 | 43.06 | 37.60 | 39.87 | 41.81 | 45.28 |
| APLLR | 32.67 | 35.34 | 35.73 | 37.76 | 33.26 | 36.87 | 39.19 | 42.62 | 36.62 | 39.89 | 42.28 | 45.78 |
| LMNN | 31.30 | 33.53 | 35.84 | 36.92 | 32.19 | 34.48 | 36.49 | 38.84 | 34.39 | 36.62 | 40.19 | 44.17 |
| SSML-DR | **35.35** | **37.53** | **39.47** | **41.38** | 37.01 | 39.46 | 42.87 | 44.59 | 39.84 | 42.69 | 45.92 | 47.25 |
| ISMLP | 34.01 | 36.52 | 38.72 | 40.29 | **37.11** | **40.38** | **43.48** | **45.81** | **40.19** | **43.58** | **46.39** | **48.41** |

### 5.5. Sensitivity Analysis

In this section, we conduct an experiment on the Cars-196 dataset to analyze the sensitivity of the proposed ISMLP on different hyper-parameters. To simplify the experiment, we keep the other parameters fixed when analyzing the current one.

Figure 4a depicts the R@1 accuracy of the proposed ISMLP with different $\lambda$ when we set $\mu = 0.001$, $p = 50$, and $m = c$, where c denotes the number of classes. One can observe that each curve has a turning point, and the fewer the amount of labeled data, the earlier the turning point appears. We gauge that this can be attributed to the utility of the entropy regularization; either an excessively large or a small $\lambda$ will lead to a biased model.

Figure 4b shows the sensitivity of ISMLP on $\mu$. We can see that ISMLP is insensitive to the change of $\mu$ to some extent. However, setting a large $\mu$ will impose the learned **R** close to the prior metric $\mathbf{I}_p$, which prevents ISMLP to learn the correlations and weights of the feature in the reduced space.

Figure 4c documents the results on $p$. Recall that $p$ is the dimensionality of the reduced space, and setting a small $p$ will lose a large amount of information of the original data. As a result, we can observe inferior results with small $p$; however, as $p$ increases, the performance becomes stable. To compromise between accuracy and computational efficiency, we set $p$ as 50 in all experiments.

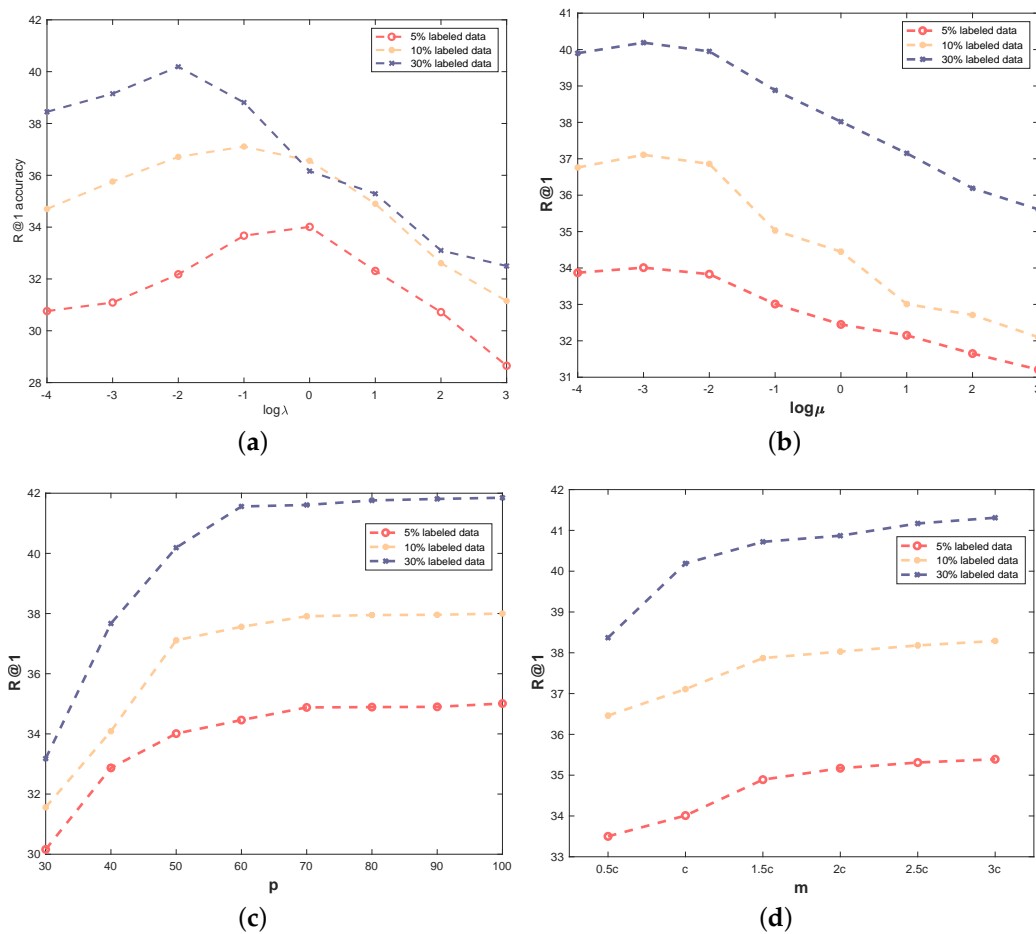

**Figure 4.** The R@1 accuracy under different parameters. We keep the other parameters fixed when analyzing the current one. (**a**) $\lambda$; (**b**) $\mu$; (**c**) Reduced dimensionality; (**d**) Number of proxy vectors.

We further conduct an experiment to test the sensitivity of ISMLP on the number of proxy vectors and document the result in Figure 4d. It has nearly the same tendency as Figure 4c; when we learn a small number of proxy vectors, instances from the other classes will be mixed up together, which undoubtedly degrades the discriminative ability of the model. Increasing the number of proxy vectors will boost the performance to some extent,

and it can help to discover the latent pattern within a class. Such an idea is also utilized in some cluster-based multi-metric learning methods [30,50]. However, allocating too many proxy vectors will cost additional computational resources.

## 6. Conclusions

In this paper, we propose an efficient information-theoretic-based semi-supervised metric learning method called ISMLP. By learning a hierarchical form of the Mahalanobis matrix as well as a set of proxy vectors, ISMLP casts the semi-supervised metric learning problem as a probability model. Importantly, the entropy regularization term is adopted to mine the rich unlabeled information. We further prove that ISMLP can be efficiently solved via the alternating direction method. Extensive experiments on five large-scale image datasets reveal that (1) the proposed probability model based on proxy vectors can accurately mine the rich information of unlabeled data, and is thus profitable for semi-supervised learning tasks; (2) ISMLP can be more efficiently trained than the semi-supervised learning methods used in the experiment; and (3) ISMLP is not sensitive to its parameters to some extent.

Despite its promising result, the proposed ISMLP assumes linear separability of the data, which is often unrealistic due to the presence of complex data structures. The kernel tricks or the deep neural networks can be utilized to extract the nonlinear features to enhance the performance. In the future, we plan to extend the proposed ISMLP to multi-model settings to deal with the multi-modal input [51–53].

**Author Contributions:** Conceptualization, P.C. and H.W.; methodology, H.W.; software, H.W.; validation, P.C. and H.W.; formal analysis, H.W.; investigation, H.W.; resources, P.C.; data curation, H.W.; writing—original draft preparation, H.W.; writing—review and editing, P.C. and H.W.; visualization, P.C.; supervision, P.C. All authors have read and agreed to the published version of the manuscript.

**Funding:** This research was funded by Dalian Science and Technology Innovation Fund 2021JJ12GX028.

**Institutional Review Board Statement:** Not applicable.

**Informed Consent Statement:** Not applicable.

**Data Availability Statement:** All datasets are publicly available.

**Conflicts of Interest:** The authors declare no conflict of interest.

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
