# Peer review of "Efficient Information-Theoretic Large-Scale Semi-Supervised Metric Learning via Proxies"

_applsci, doi:10.3390/app13158993_

Round 1
Reviewer 1 Report
This paper proposed a novel method entitled Information-theoretic large-scale Semi-supervised Metric Learning via Proxies (ISMLP) for large dataset training without incurring too much time. This topic is interesting and the related algorithms have been well demonstrated. However, in the experiment parts, the authors had better provides more explanation to the experimental results to further illustrate the difference between the proposed methods and the other existing similar algorithms. Also, if the authors can provide the limitations of the proposed model and have a brief discussion, that would be better.
There are concerns regarding the English grammar and clarity of the presented work. This paper needs some modification of certain sentences to make them correct.
Reviewer 2 Report
1. The article proposes a novel semi-supervised metric learning method called ISMLP. Can you explain in more detail how ISMLP differs from existing methods and what makes it unique?
2. The article claims that ISMLP achieves superior performance compared to other semi-supervised metric learning methods. What are the specific metrics used to evaluate this claim, and are there any potential limitations or biases in the evaluation process?
3. The introduction mentions that existing semi-supervised metric learning methods often rely on manifold assumptions. How does ISMLP address this issue, and what evidence or experiments are provided to support its effectiveness in handling unlabeled data?
4. The article emphasizes the linear time complexity of ISMLP as a significant advantage for large-scale datasets. Can you elaborate on how this linear complexity is achieved and whether there are any trade-offs in terms of model performance or other aspects?
5. In Section 3, the article introduces the concept of proxy vectors and their role in ISMLP. How does using proxy vectors instead of instance-instance distances affect the overall performance and efficiency of ISMLP?
6. The article mentions that ISMLP incorporates entropy regularization to preserve discriminative information in the unlabeled data. Can you explain how entropy regularization works and whether there are any potential drawbacks or challenges in using this technique?
7. In the experiments, the article evaluates ISMLP's performance on various datasets. How diverse are the datasets used, and are there any concerns regarding the generalizability of the results to other types of datasets?
8. The article includes comparisons with several other semi-supervised metric learning methods. Were the compared methods selected based on specific criteria, and are there any other relevant state-of-the-art methods that could have been included in the comparison?
9. The proposed ISMLP method relies on hyperparameters such as λ, µ, p, and the number of proxy vectors. How sensitive is the performance of ISMLP to changes in these hyperparameters, and how were the specific values for these hyperparameters chosen in the experiments?
10. The article mentions that the proxy vectors used in ISMLP are more stable than those used in other methods like SERAPH. Can you provide a more in-depth analysis of this stability and how it impacts the robustness of the model to noisy data?
11. In the conclusion, the article suggests possible future directions for ISMLP. What are some potential extensions or improvements that can be made to enhance the performance or applicability of ISMLP in real-world scenarios?
12. How does the proposed ISMLP method compare to deep learning-based metric learning methods, which have gained popularity in recent years? Are there any specific advantages or disadvantages of ISMLP in comparison to deep learning approaches?
13. Figures 1 and 4 needs improvement in terms of quality.
14. In Tables 2,3 and 4, the font is so small, you need to find a way how to enlarge the font to make it readable.
Reviewer 3 Report
· On page 10 “Evaluation protocol: The default split of the train/test is adopted” the authors should clarify what is the default split?!
· In Section (4.1 Time Complexity of ISMLP): This section is not supported by time analysis or/and time comparison with other method. This section should be extended
· In table 5, for the 5% labeled data the proposed method (ISMLP) achieve similar results to the method of SERAPH [27], more in-depth analysis is required to show the pros of the proposed method over the SERAPH method.
A submision of a revised version of the manuscript is required
Minor editing of English language required
Round 2
Reviewer 3 Report
The authors have addressed my previous concerns. No further comments.
English language is fine